# Landau–de Gennes Model for the Isotropic Phase of Nematogens: The Experimental Evidence Challenge

**DOI:** 10.3390/ijms26209849

**Published:** 2025-10-10

**Authors:** Sylwester J. Rzoska, Aleksandra Drozd-Rzoska, Tushar Rajivanshi

**Affiliations:** Institute of High Pressure Physics of the Polish Academy of Sciences, ul. Sokołowska 29/37, 01-142 Warsaw, Poland; sylwester.rzoska@unipress.waw.pl (S.J.R.); tusharrajvanshi3@gmail.com (T.R.)

**Keywords:** Landau–de Gennes model, isotropic liquid phase, isotropic–nematic transition, prenematic fluctuations, dielectric constant, nonlinear dielectric effect, molecular structure

## Abstract

The Landau–de Gennes model is one of the most significant fundamental frameworks in *The Physics of Liquid Crystals* and Soft Matter Physics. It is validated by the universal parameterisation of the Cotton–Mouton effect, the Kerr effect, and light scattering in the isotropic phase of nematogens. However, as early as 1974, de Gennes identified the first two puzzling problems of this model. Over the following decades, this list has expanded. This report presents the first comprehensive analysis of these issues, with the explicit experimental reference. It focuses on the hardly coherently discussed pretransitional changes in the dielectric constant and the extension in a strong electric field, specifically the nonlinear dielectric effect (NDE). Notably, there are uniquely different pretransitional forms of pretransitional effects, depending on molecular structural features such as permanent dipole moment loci or a steric hindrance. It is tested for 5CB, 5*CB, and MBBA: nematogenic liquid crystalline materials that differ in the above features. The obtained specific pretransitional effects and the evidence for the essential importance of the interplay between observation and pretransition fluctuations time scales led to a new coherent, model-based explanation of all the discussed problems, which cannot be explained within the canonical Landau–de Gennes model.

## 1. Introduction

The isotropic phase of rod-like nematic liquid crystals (LCs) is a gateway to liquid crystal (LC) mesophases whose extraordinary properties inspired the research validating the Landau–de Gennes model, one of the crucial concepts in *The Physics of Liquid Crystals* [1,2,3,4,5,6,7]. Its significance includes the *Soft Matter* category [8,9,10,11,12], comprising liquid crystals, near-critical systems, polymers, colloids, micellar systems, supercooling glass formers, …. and model foods or biosystems [9,10,11,12,13,14,15]. It demonstrates the interdisciplinary nature of the field, which brings together physicists, chemists, and biologists [1,2,3,4,5,6,7,8,9,10,11,12,13,14,15]. In recent years, soft matter models have also entered the fields of socio-economics [16] and topology-governed systems [17]. Despite qualitative differences, these systems share common scaling patterns. The isotropic phase of nematogens is also important in *Critical Phenomena and Phase Transitions Physics* due to a melting/freezing unusual discontinuous phase transition related to only one element of symmetry [18,19,20,21]. The isotropic phase of rod-like nematogenic LC has also proven to be significant for studying previtreous properties in glass-forming systems, i.e., for the *Glass Transition Physics* [22,23,24,25,26].

A unique significance of the isotropic phase of the nematogens also stems from the fact that it constitutes an experimental model system, facilitating the following cognitive feedback process: experiment ←→ (theory, simulation).

The Landau–de Gennes (LdG) model is a key reference in all of the aforementioned fields. Pretransitional effects studies in the isotropic liquid phase of rod-like nematic LC constitute the canonical reference for LdG model validation, most often recalled in monographs [2,3,4,5,6,7,8,9,10,12,19,20]. However, when compared with experimental results, several puzzling issues emerge. This report presents this peculiar experimental evidence and proposes coherent model explanations of emerging challenges.

Five decades ago, Pierre Gilles de Gennes published seminal papers discussing the anomalous increase in the Cotton–Mouton effect (CME) in the isotropic liquid phase of rod-like liquid crystalline (LC) compounds, as they approach the isotropic–nematic (I-N) transition [27,28]. To explain the surprisingly simple and common pattern of changes, de Gennes considered the expansion of the Landau model free energy (F) using the uniaxial quadrupolar order parameter [27,28]:(1)$$F\left(T\right)=F_{o}+\frac{1}{2}AQ_{ij}Q_{ji}+\frac{1}{3}BQ_{ij}Q_{jk}Q_{ki}+0\left(Q^{4}\right)-\frac{1}{2}GQ_{ij}H_{i}H_{j}+\ldots$$

Substituting the tensorial uniaxial order parameter $Q_{ij}=S\langle n_{i}n_{j}-\delta_{ij}/3\rangle$, where $\left(n_{i},\;n_{j}\right)$ refers to the component of the director vector with ‘nematic’ $\vec{n}$ and $\vec{-n}$ and S is the scalar metric of ordering, de Gennes obtained the expansion with the scalar metric [1,2,27,28]:(2)$$F\left(T\right)=F_{o}+\frac{1}{2}AS^{2}+\frac{1}{3}BS^{3}+CS^{4}-\frac{1}{2}GSH_{i}H_{j}+\ldots$$

De Gennes introduced the extrapolated temperature of a hypothetical continuous phase transition in order to take into account the long-range pre-transitional effect associated with discontinuous phase transitions: $T^{*}=T_{I-N}-\Delta T^{*}$, where $T_{I-N}$ is the I-N ‘clearing’ temperature and $\Delta T^{*}$ is the I-N discontinuity metric. Consequently, the amplitude of the second-order term in the above relation is given by $A=a\left(T-T^{*}\right)$. The last term in the above equations reflects the impact of the external field. Refs. [27,28] recalled Cotton–Mouton effect (CME: refractive index birefringence induced by a strong magnetic field).

In subsequent years, apart from the CME [27,28,29], such modelling was employed for the Optical Kerr effect (*OKE*) [30,31,32], Electro-Optic Kerr effect (*EKE*) [32,33,34], Rayleigh light scattering (*LS*, $I_{R}$) [35], turbidity (θ) [36], compressibility ($\chi_{T}$) [37], and nonlinear dielectric effect (*NDE*, $\Phi_{E}$) [38]. For each of these methods, the same temperature pattern for the strong and long-range pretransitional effect was reported [1,2,3,4,19,20,29,30,31,32,33,34,35,36,37,38]:(3)$$CME,KE,\;{\Phi_{E},\;I}_{R},\theta ,\;\chi_{T}=\frac{A_{M}}{T-T^{*}}\;\;\;\Rightarrow \;\;\;\;\;\;\;\;\;\;\;{\Rightarrow \;CME}^{-1},{KE}^{-1},\;I_{R}^{-1},\;\theta^{-1}\;\;\chi_{T}^{-1},\Phi_{E}^{-1}=A^{-1}T-\;A^{-1}T^{*}=aT-b$$
where $A_{M}$ is the amplitude related to the given method; parameters a,b=const.

In the basic monographs for *Liquid Crystals Physics*, the linear changes in CME, KE, or LS reciprocals are presented as the crucial validation of the Landau–de Gennes (LdG) model, as well as a more general mean-field description validation in the isotropic liquid phase [1,2,3,4,5,6,7].

The grand success of the LdG model was a significant contribution to honouring Pierre G. de Gennes’ Nobel Prize in 1991 [39].

However, already in the first edition of the grand monograph *The Physics of Liquid Crystals* (1974, [1]), de Gennes pointed out some puzzling problems of the model:
**(Q1)** ‘For the majority of second-order phase transitions the Landau model approximation does not occur (magnetic systems, superfluid Helium, …) showing for response functions the behaviour $\propto {\left(T-T_{C}\right)}^{-\gamma }$, with the exponent 1.25<γ<1.45, but not $\propto {\left(T-T_{C}\right)}^{-1}$ ‘ [1].

It becomes even more puzzling when including later results for Electro-Optic Kerr effect (EKE) and NDE upon approaching the critical consolute point, with anomalies described by the exponents $\psi_{EKE}\approx 0.85$ and $\psi_{NDE}\approx 0.39$, respectively [40,41].
**(Q2)** ‘Why is the temperature singular $T^{*}$ is so close to $T_{I-N}$, i.e., $\left(T^{*}-T_{I-N}\right)/T_{IN}\sim 2\times 10^{-3}$ ?’ [1].

It means that the experimentally detected discontinuity via properties encountered in Equation (3) is equal to ${\Delta T}^{*}=T_{I-N}-T^{*}=1-2K$ [1,4,19,21,29,30,31,32,33,34,35,36,37,38]. Notably, the mean-field analysis via basic Landau–de Gennes or Maier–Saupe models gives ${\Delta T}^{*}\sim 30K$ [1,4,42,43,44].

The results obtained in subsequent decades showed that this list can be significantly longer, namely, the following:
**(Q3)** For CME, KE, and $I_{R}$, pretransitional changes and characteristic deviations from the LdG Equation (3) appear in the immediate vicinity $T_{I-N}$. They are commented only heuristically via the impact of not-defined ‘cybotactic groups’ [3,4,31,32,45,46,47,48,49,50,51,52].**(Q4)** Heat capacity/specific heat shows the pretransitional anomaly $c_{p}\propto {\left|T-T_{C}\right|}^{-\alpha }$, with the exponent *α* = 1/2, in the isotropic liquid and in the nematic phases [4,19,47,48,49,50,51]. The basic LdG model yields in the isotropic liquid phase *α* = 0 [4,5,6,53,54,55,56]**(Q5)** The puzzle regarding the explanation of the pretransitional anomaly of the dielectric constant $\varepsilon \propto {\left(T-T^{*}\right)}^{1-\alpha }$ existed [57,58,59,60,61,62,63,64]. It includes the explicit dependence on the molecular structure [63,64].**(Q6)** Nonlinear dielectric effect (NDE) is the strong electric field-related extension of dielectric constant [65,66,67]:(4)$$\varepsilon \;\left(E\right)=\varepsilon +{\Delta \varepsilon }^{E}E^{2}+\cdots \;\;\;\;\;\;\;\;\to \;\;\;\;\;\;\;\;\;\;NDE=\frac{{\Delta \varepsilon }^{E}}{E^{2}}=\frac{\varepsilon \left(E\right)-\varepsilon }{E^{2}}$$

Generally, the infinite rise in NDE upon approaching a critical point is expected: ${\Delta \varepsilon }^{E}/E^{2}\to \;+\infty ,\;T\;\to {\;T}_{C}$. However, for some exceptional LC materials, the ‘negative’ pretransitional anomaly was found: ${\Delta \varepsilon }^{E}/E^{2}\;\Rightarrow \;-\infty ,\;T\;\to {\;T}_{C}$ [68].

The unusual, universalistic pretransition changes described by Equation (3) served as the inspiration and origin for the Landau–de Gennes model. Hence, significant experimental problems may raise both concerns and a call for clarification.

Notably, the Landau–de Gennes model has remained the subject of intensive theoretical research for decades, aimed at testing its peculiarities and the limits of its applicability [1,2,3,4,5,6,7,8,9,15,69,70,71,72,73]. However, this report refers to the puzzling experimental evidence (**Q1**–**Q6**) that has not been explained so far, and is essential for the ultimate validation of the model.

To address this challenge, focused studies of the dielectric constant and NDE are essential. These results are presented below and are followed by model considerations in the Section 3. They comment on and explain the puzzling experimental features recalled in questions **Q1**–**Q6**.

## 2. Results

The results presented below are for three nematogenic rod-like liquid crystalline compounds, with the following mesomorphism:(i)4-cyano-4-n-pentylbiphenyl (pentylcyanobiphenyl, 5CB), with the following mesomorphism [4,52]:$$Solid\;Crystal\;\;\left(Cr\right)-\;286.6\;K-Nematic\;\left(N\right)-307.6\;K-Isotropic\;\;(I)$$(ii)4-(2-methylbutyl)-4′-cyanobiphenyl (isopentylcyanobiphenyl, 5*CB [69,74,75]: $$Solid\;Glass\;\left(G\right)-220\;K-Chiral\;Nematic\;\left(N^{*}\right)-250\;K-Isotropic\;(I)$$(iii)p-methoxy-benzylidene p-n-butyl-anilline, MBBA [4,21,52]: $$Solid\;Crystal\;\;\left(Cr\right)-\;292\;K-Nematic\;\left(N\right)-318.5\;K-Isotropic\;(I)$$

5CB and 5*CB share the same chemical formula C_18_H_19_N, but they differ slightly in molecular structure, as depicted in Figure 1 and Figure 2 below. 5CB essentially crystallizes on cooling. Supercooling is only possible when fast cooling is matched with a specially prepared sample [74]. 5*CB can be supercooled at an arbitrary cooling rate down to the glass temperature that occurs in the chiral nematic phase at $T_{g}\approx 220K$ [68,76,77]. Both 5CB and 5*CB are associated with the permanent dipole moment μ≈5.0D [4,15], approximately parallel to the long molecular (cyanobiphenyl) part of molecules. It leads to the positive anisotropy of dielectric permittivity: $\Delta \varepsilon =\varepsilon_{\left|\right|}\;-\;\varepsilon_{⊥}\approx 11$ for 5CB [4].

For MBBA, the permanent dipole moment is almost perpendicular to the long molecular axis, μ=1.983D, and it is located at the angle β≈86° with respect to the long molecular axis. It leads to the negative anisotropy of dielectric permittivity, $\Delta \varepsilon =\varepsilon_{\left|\right|}\;-\;\varepsilon_{⊥}\approx -0.54$ [75]. The tested compounds were purchased from Fluka and carefully degassed before the experiments.

In conclusion, the tested compounds are characterized by qualitatively different positions of the permanent dipole moment (5CB and MBBA) and the presence of structural steric hindrance (5*CB). These factors have a significant impact on the form of pretransitional effects in the isotropic liquid phase, as noted in the literature and as presented in the results below. This is a unique situation, as the standard properties studied in the isotropic phase, such as compressibility (usually tested via KE or IR measurements) and heat capacity (specific heat), consistently exhibit the same pretransitional patterns. This includes 5CB, 5*CB, and MBBA, tested in the given report.

Figure 1 and Figure 2 present examples of registered spectra for the real part of dielectric permittivity in 5CB and 5*CB, focusing on features relevant to the physical properties discussed in the given report. High-resolution spectra for MBBA have only recently been reported in ref. [64] Both figures show the same pattern, with a long horizontal domain where the frequency shift does not change the value of $\varepsilon^{\prime }\left(f\right)$. This is the static domain, defining the dielectric constant, i.e., ${\varepsilon =\varepsilon }^{\prime }\left(f\right)$ [65].

When increasing the frequency above the static domain $\varepsilon^{\prime }\left(f\right)$ drops, which is the hallmark that the long molecular axis gradually ceases to follow changes in the electric field. This is the onset of the dynamic domain used for testing relaxation processes, which are not the focus of the given report. On lowering the frequency below the static domain, values of $\varepsilon^{\prime }\left(f\right)$ ‘dramatically’ rises. It is the low-frequency region, and the rise is generally linked to ‘ionic impurities’, whose impact can be detected for low enough excitation frequencies [60,61,65]. For the authors, an alternative explanation via translational shifts in the basic molecules can yield a similar registered output [64].

Generally, experimental studies of dielectric constant temperature evolution are related to the real part of dielectric permittivity scans at a selected, constant frequency, most often in the range from 10 kHz to 100 kHz, and are carried out [4,21,56,57,58,59,60,61,62,63,64]. Such a scenario can be successful for 5CB and also for MBBA, with a similar $\varepsilon^{\prime }\left(f\right)$ spectrum. However, in the isotropic phase of 5*CB, the static domain strongly shifts on cooling (see Figure 2), which is related to the enormous temperature range possible for testing and approaching the glass temperature. It causes the above constant frequency scenario to fail. For the 5*CB dielectric constant studies, the focus should be on the middle of the static domain, shifted towards lower frequencies upon cooling. As visible in Figure 2, for 5*CB, the static domain extends from ca ~3 kHz to well below 1 Hz just above the clearing temperature. For 5CB and also for MBBA, it is almost always related to the frequency range 2 kHz<f<7 MHz.

### 2.1. Pretransitional Behaviour of Dielectric Constant

Figure 3 shows dielectric constant changes in the isotropic liquid phase of MBBA, up to $\sim T^{C}+65K$. There is an explicit linear behaviour, with no hallmarks of a pretransitional effect for $T^{C}\leftarrow T$, i.e., upon approaching the I-N transition. Nevertheless, the derivative analysis of high-resolution data in ref. [64] revealed weak hallmarks of the pretransitional behaviour for $d\varepsilon \left(T\right)/dT$ tests.

Figure 4 presents changes in the dielectric constant in the isotropic liquid phase of 5CB and 5*CB, using the normalized scales. The plot shows the explicit manifestations of pretransitional effects, namely(5)$$d\varepsilon \left(T\right)/dT>0\;\leftarrow \;T^{C}\;\leftarrow \;T\;\leftarrow \;d\varepsilon \left(T\right)/dT<0$$
where the clearing temperature $T^{C}$ is related to I-N transition for 5CB and I-N* for 5*CB.

Such pretransitional behaviour is particularly evident in the inset in Figure 5, for the focused behaviour in the close vicinity of $T^{C}$.

The curves portraying pretransitional changes are related to the following relation [56,57,58,59,60]:(6)$$\varepsilon \left(T\right)=\varepsilon^{*}+A_{\varepsilon }\left(T-T^{*}\right)+B_{\varepsilon }{\left(T-T^{*}\right)}^{\varphi }$$
where $\left(\varepsilon^{*},T^{*}\right)$ is for parameters describing the hypothetical continuous phase transition, at the tested temperatures $T>T^{C}=T^{*}+{\Delta T}^{*}$; the exponent is related to the one describing the mean-field related heat capacity anomaly, namely *φ* = 1 − *α* = 1/2 [57,58].

For 5CB and 5*CB, dielectric constant tests enable determining the values of the discontinuity metric: for both compounds within the limit of the experimental error, $\Delta T^{*}\approx 1.3_{5}\;K$. Worth stressing is the range of the extreme range of pretransitional effects and significantly lesser ‘strength’ of the pretransitional effect for 5*CB in comparison with 5CB. It is reflected by the relation of amplitudes $B_{\varepsilon }\left(5CB\right)/B_{\varepsilon }\left(5^{*}CB\right)\approx 1.8$, as results from the data given in Table 1 and visible in Figure 4.

There are essential problems with the direct derivation of the relation for $\varepsilon \left(T\right)$ directly from the Landau–de Gennes model, and includes the molecular structure impact.

To portray $\varepsilon \left(T\right)$ anomaly in the isotropic phase, Drozd-Rzoska [58,63] recalled Mistura’s [78] model reasoning for precritical changes in the dielectric constant in the homogeneous phase above the critical consolute point (CCP) and gas–liquid–critical point (GLCP). Mistura indicated the link to the critical behaviour of the specific heat/heat capacity $C_{V}$ [78]:(7)$$\frac{d\varepsilon \left(T\right)}{dT}\propto C_{V}\left(T\right)\propto {\left(T-T_{C}\right)}^{-\alpha }$$
where $T_{C}$ denotes the critical temperature.

For systems with CCP and GLCP, the critical exponent *α* ≈ 0.115. It is the value for the d=3 (space dimensionality) and n=3 (order parameter dimensionality) universality class [79]. The Mistura model concept [78] served as the essential reference for Sengers et al.’s [80] model analysis, leading to an equation that portrays ε(T) changes upon approaching CCP and GLCP, solving theoretically the challenge that has existed since 1934 [81]. The functional form of the model relation derived by Sengers et al. [80] agrees with Equation (6), although it additionally includes critical corrections-to-scaling terms, important as we move away from the CCP, starting only from 2 K above $T_{C}$ [82,83].

Drozd-Rzoska [58,63] noted that Mistura’s model [78] analysis does not explicitly refer to systems associated with the critical universality class (d=3, n=1), but addresses generally a supercritical liquid (above the critical point) with the dominant impact of pretransitional, critical fluctuations. For the isotropic phase of nematogens, prenematic fluctuations are observed, and the critical temperature can be linked to the extrapolated singular temperature $T^{*}$, for instance, of a pseudospinodal type [84]. Drozd-Rzoska further recalled the mean-field character of the isotropic phase, leading to the critical exponent *α* = 1/2 for the specific heat anomaly. Subsequently, the integration of Equation (7) can directly yield Equation (6) when additionally taking into account the ‘background’ term reflecting linear temperature changes in the nondipolar polarization component [58,63]. Finally, for the mean-field description, correction-to-scaling terms, appearing in Sengers et al.’s [80] model, are inherently absent.

In 1983, Thoen and Menu [57] carried out high-resolution dielectric constant measurements in the isotropic liquid phase on n-octyl-cyanobiphenyl, a rod-like LC compound with Isotropic liquid − Nematic − Smectic A − Crystal mesomorphism. They demonstrated a fair portrayal using Equation (6), with the heuristic explanation indicating some informal similarity to the behaviour in the homogeneous phase as it approaches the critical consolute point. The Drozd-Rzoska model [58,63] provided a formal justification for this preliminary result.

### 2.2. Pretransitional Behaviour of Nonlinear Dielectric Effect

This section presents the essential evidence for the dielectric constant counterpart of a strong electric field—the nonlinear dielectric effect (NDE) [65,66,67], defined by Equation (4). The pattern of NDE temperature changes in the isotropic phase of nematogenic LC compounds can be directly derived from the Landau–de Gennes model, similarly to other properties encountered in Equation (3). Recalling the analysis from refs. [85,86], one obtains the following:(8)$$\Phi_{E}\left(T\right)=\frac{{\Delta \varepsilon }^{E}}{E^{2}}=\frac{2}{3a\varepsilon_{0}}F\frac{{\Delta \varepsilon }^{0}{\Delta \varepsilon }^{f}}{T-T^{*}}=C\frac{{\Delta \varepsilon }^{0}{\Delta \varepsilon }^{f}}{T-T^{*}}=\frac{A_{NDE}}{T-T^{*}}$$

Consequently, the NDE reciprocal should follow a linear pattern, as indicated in Equation (3). It essentially simplifies the analysis of experimental results.

In the above relation, $C=2F/3a\varepsilon_{0}=const$. The coefficient ‘*a*’ is related to the second-order amplitude in Equation (2) and defines the amplitude of compressibility, or, alternatively, orders parameter-related susceptibility $\chi_{T}=\chi_{0}{\left(T-T^{*}\right)}^{-\left(\gamma =1\right)}$ with $\chi_{0}=a^{-1}$; F is the local field-related factor reflecting the fact that the electric field acting on a molecule within a dielectric material differs from the externally applied one; ${\Delta \varepsilon }^{0}$ and ${\Delta \varepsilon }^{f}$ are metrics of molecular anisotropies of ‘dielectric constant’ for the hypothetical zero-frequency limit and the applied measurement frequency. This is related to the action of the DC strong electric field and the weak scanning field, respectively.

Recalling experimental conditions, values of molecular anisotropies in Equation (8) should be associated with the real part of dielectric anisotropy determined for frequencies in the static domain for LC samples in the nematic phase perfectly oriented by a strong magnetic field in directions perpendicular and parallel to the scanning electric field. Hence, ${\Delta \varepsilon }^{0}=\Delta \varepsilon =\varepsilon_{\left|\right|}-\;\varepsilon_{⊥}$ in the static domain and ${\Delta \varepsilon }^{f}={\left({\Delta \varepsilon }^{\prime }\right)}^{f}$, i.e., it can also cover frequencies beyond the static domain.

Figure 5 shows the NDE pretransitional rise in the isotropic phase of 5CB and MBBA for two scanning frequencies of the weak electric field. For MBBA, the pretransitional effect is ca. 50× weaker than for 5CB: it means it is of the order $\left(10^{-18}-10^{-17\;}m^{2}V^{-2}\right)$ for MBBA and $\left(10^{-16}\;m^{2}V^{-2}\right)$ for 5CB.

This difference does not influence the resolution of registered NDE values. For the given apparatus concept, one can detect even the lowest possible contribution to NDE from statistical polarization fluctuations in the nondipolar liquid, of the order $\left(10^{-19}-10^{-20\;}m^{2}V^{-2}\right)$ [65,66]. The inset in Figure 5 is for the reciprocal of NDE changes from the main part of the plot, focused on the visual validation of the LdG model-related temperature pattern (Equations (3) and (8)). Notable is the range of such descriptions extending even above $\sim T^{C}+50\;K$.

For MBBA, the shift in the scanning frequency does not influence the phenomenon. However, such an impact significantly manifests in 5CB for the scanning frequency f=5 MHz. It should be stressed that both frequencies are related to the static domain of the dielectric constant, as shown in Figure 1 for 5CB and in ref. [64] for MBBA.

Figure 6 presents the pretransitional behaviour in the isotropic liquid phase of glass-forming 5*CB. The striking feature is the anomalous pretransitional effect, which tends toward negative values. The impact of the scanning frequency is also visible. The lower of the tested frequencies corresponds to the static domain of the dielectric constant, as shown in Figure 2. Generally, for obtaining the ‘negative’ pretransitional anomaly for NDE within the Landau–de Gennes model, opposite signs of anisotropies ${\Delta \varepsilon }^{0}$ and ${\Delta \varepsilon }^{f}$ are required in the context of Equation (8). However, it is not possible to obtain these single-molecule-related properties for 5*CB. Nevertheless, the question regarding the parameterization arises.

The standard way of testing NDE, KE, $I_{R}$ ….pretransitional patterns is the ‘reciprocal plot’, as shown in the inset in Figure 5. However, for 5*CB, the crossover $NDE<0\leftarrow \underline{NDE=0}\;\leftarrow \;NDE>0$ excludes such analysis.

To overcome this problem, the following transformation of experimental data has been carried out:(9)$$\Phi_{E}\left(T\right)=\frac{A_{NDE}}{T-T^{*}}\;\;\Rightarrow \;\frac{d\Phi_{E}}{dT}=\frac{A_{NDE}}{{\left(T-T^{*}\right)}^{2}\;}\;\Rightarrow \;\Rightarrow \;{\left(\frac{d\Phi_{E}}{dT}\right)}^{-2}\;=\;A_{NDE}^{-1}T-A_{NDE}^{-1}T^{*}\;=\;a_{\Phi }T\;-\;b_{\Phi }$$
where constant parameters are related to the following linear behaviour: $a_{\Phi }=A_{NDE}^{-1}$ and $b_{\Phi }=A_{NDE}^{-1}T^{*}$.

The inset in Figure 6 shows the results of such data transformation, based on Equation (9). The fair linear behaviour validates the mean-field type evolution $\Phi_{E}\left(T\right)=1/\left(T-T^{*}\right)$, with the negative amplitude $A_{NDE}<0$.

As for the behaviour well remote from the clearing temperature, one may expect that the influence of pretransitional fluctuation diminishes. The form of the temperature evolution in this domain, shown by the brown line, allows us to recall the classic Piekara model for NDE changes in molecular liquids with the tendency to pair dipole–dipole coupling [65,87]:(10)$${\left(\frac{{\Delta \varepsilon }^{E}}{E^{2}}\right)}_{coupl.}=-F\frac{N\mu^{4}}{k_{B}T^{3}}R_{P}$$
where N is for the number of permanent dipole moments in a unit volume, μ denotes the permanent dipole moment metric, $k_{B}$ is the Boltzmann factor, and $R_{P}\left(T\right)$ denotes Piekara’s factor describing the possible dipole–dipole coupling and for the antiparallel arrangement $R_{P}<0$; F is the local electric field factor.

Originally, Equation (8) was introduced to explain the positive NDE value in nitrobenzene, with the explicit antiparallel dipole–dipole local arrangements [65,87].

## 3. Discussion

This section addresses the experimental challenges presented in questions Q1–Q6, with emphasis on the experimental results discussed above. Section 3.1 provides some cognitive background comments for the *Physics of Critical Phenomena*, which are significant for the subsequent model discussion.

### 3.1. Critical Fluctuations and Mean-Field Behaviour

The isotropic–nematic transition in rod-like molecular systems exhibits unique features of an experimental model system, i.e., a system that can be relatively simply mirrored by a conceptual/theoretical model, offering exceptional experiment−theory cognitive feedback analysis [4,19,20,21,22,23,24,25,26,27,28,29,30,31,32,33,34,35,36,42,43,44,45,46,47,48,49,50,51].

For this path, of particular importance is the molecular field interaction model for uniaxial rods considered by Onsager in 1946 [88], later developed by Flory [89] and followers [90,91,92,93,94]. These theoretical studies, later supplemented by simulations, showed that a specific I-N solidification process is associated only with uniaxial orientation, i.e., a single element of symmetry, freezing. The discontinuous character, with step changes in relevant physical properties, can be considered a generic feature of such systems, as indicated already by Onsager [88].

It is essentially a comment on the discontinuous character of the I-N transition in rod-like LC systems. However, a question arises as to why this discontinuity is so small, as pointed out by de Gennes in his **Q2** question.

It seems that a simple heuristic answer might be the ‘freezing’ limited to only one element of symmetry. Two further essential experimental facts should also be considered: (***i***) the extremely long-range pretransitional effects, which are generally absent for standard discontinuous melting/freezing transitions, and (***ii***) the mean-field nature of these pretransitional effects.

Regarding the first issue, for weakly discontinuous phase transitions, the nature of the phenomenon often causes it to be associated with a critical-like anomaly ‘hidden’ just below the transition. An example is pseudospinodal behaviour in the close surroundings of a critical point [84]. Such a ‘pseudospinodal’ picture can resemble the behaviour observed for the I-N phase transition [84,95,96].

Generally, pretransitional effects are caused by the appearance of multimolecular pretransitional fluctuations, which exhibit locally features of the next, approaching phase. For the given case, they are prenematic fluctuations. The size (correlation length, ξ) and lifetime ($\tau_{fl}$) of pretransitional fluctuations critically increase upon approaching the singular temperature, critical $T_{C}$, or the pseudospinodal $T_{Sp}$ temperatures [79,84]:(11)$$\xi \left(T\right)\propto \frac{1}{{\left(T-T_{C,\;Sp}\right)}^{\nu }}\;\;\;,\;\;\;\tau_{fl.}\left(T\right)\propto \frac{1}{{\left(T-T_{C,\;Sp}\right)}^{z\nu }}$$
where for the cases of GLC and CCP, the exponent ν=0.625…, and the dynamic exponent z=2 for the conserved and z=3 for the non-conserved order parameter.

For the I-N transition, one can assume the following: $T_{C}\;\to \;T^{*}$: ν=1/2 and zν=1 [81].

Notably, values of critical exponents are described by space (d) and order parameter (n) dimensionalities. Hence, all near-critical systems can be arranged in (d,n) universality classes [79]. Remote from the critical/singular temperature, the correlation length of critical fluctuations reduces to as few as two, or even fewer, molecules. It is matched with their extremely short lifetime. Consequently, the impact of standard intermolecular interactions may become dominant again. This crossover is described by the Ginzburg criterion [97] in the cases of GLC and CCP. It is also the boundary between the non-classical description and the ‘classic’ mean-field type behaviour. However, pretransitional fluctuations remain relevant because the system’s properties are still described in relation to the critical/singular point. It is often emphasized that in the classic domain, the long-range intermolecular interactions, in comparison to critical fluctuations, cause the influence of the latter to be averaged out [85,86]. However, there are systems where a mean-field description occurs in the presence of large multi-element pretransitional fluctuations. Ferroelectrics in the paraelectric phase can be a canonical example. They are associated with long-range Coulomb interactions, as well as a dominance of uniaxiality, which appears to be essentially important for the appearance of the mean-field behaviour [98].

In practice, the ‘infinite’ range of interactions means the possibility of interacting with a larger number of molecules (or species) than is possible via a direct, ‘geometric’ environment of the same type of molecules. This can be achieved by a sufficiently high effective dimension of space. For a single, isolated critical point, this is related to d≥4, with the inherent mean-field description. The related critical type is universal for any system, regardless of its microscopic features. It is expressed by critical exponents: for compressibility *γ* = 1, correlation length ν=1/2, and order parameter β=1/2. For the specific heat (heat capacity), *α* = 1/2 in the low-temperature phase (below $T_{C}$) and *α* = 0 (no anomaly) in the high-temperature phase [19,98].

Another type of mean-field behaviour and pretransitional effects appears in the vicinity of the tricritical point (TCP) where three lines of critical points intersect. In some systems, a simplified symmetric TCP phenomenon can appear. It manifests via a smooth crossover from the line of discontinuous to the line of continuous phase transitions [99,100]. It was noted for some magnetic systems or phase transitions associated with the superfluid behaviour [99,100]. For TCP, the border dimensionality reduces to d=3, and the pretransitional effects are coupled with the following values of exponents: *γ* = 1, ν=1/2, β=1/4, and *α* = 1/2, both above and below the TCP singular temperature [100,101,102,103,104,105].

### 3.2. I-N Transition Mean-Field Nature

The evidence for universalistic long-range anomalous changes in the Cotton–Mouton effect, Kerr effect, and Rayleigh light scattering, obtained at the turn of the 1960s and 1970s (see Equation (3)) [27,28,29,30,31,32,33,34,35,36,37], has become the crucial evidence for validating the mean-field nature of the I-N transition and the isotropic phase. The hallmark feature is the exponent *γ* = 1 for describing critical changes in compressibility, linked to properties recalled in Equation (3). It should be noted that at that time, only Rayleigh light scattering could be explicitly linked theoretically to compressibility. The model relationship between the Kerr effect, NDE, and a simple extension to CMEs was demonstrated only in 1993 [40].

The behaviour of these properties can be phenomenologically described by the basic LdG model formulation, as defined in Equations (1) and (2). However, it also yields the exponent *α* = 0 in the high-temperature phase, which means there is no anomaly for the specific heat and related properties in the isotropic phase, and is in explicit disagreement with the experimental evidence for the specific heat [4,19,53,54,55,56] or dielectric constant (see the Section 2).

The problem was solved by implementing the more general Landau & Ginzburg model expansion, which takes into account the impact of supplementary factors, particularly reflecting weak local density fluctuations [4,19,42,43,44]. Nevertheless, it introduces additional parameters to relations describing pretransitional changes, making the fitting of experimental data tricky. To reflect the collective behaviour expressed by Equation (11), the gradient term for local changes has been included. This is the extended Landau–Ginzburg–de Gennes (L-G-dG) model [4,19].

The agreement with the specific heat can also be reached by extending the Landau–de Gennes expansion up to the term proportional to ${\propto S}^{6}$ in Equation (2), which implies the near-tricritical nature of the I-N transition [4,19,101,102,103,104,105] rather than TCP pseudospinodal-type behaviour. It naturally yields the required form of the pretransitional anomaly for the specific heat/heat capacity.

The discussion regarding the basic mean-field vs. TCP-type origin of the I-N transition has a long history [4,19,50,101,102,103,104,105], and only unambiguous experimental results can be decisive. Ultimate estimations of the order parameter exponent value in the nematic phase can be essential since the value β = ¼ for TCP and β = ½ for the basic mean field [19,79]. However, matching experimental data, most often based on refractive index or dielectric constant measurements, appears to be a challenging task due to the discontinuity of the I-N phase transition, i.e., experimental data are available only well beyond the singular temperature. These biassing factors can be minimized by the linearized, distortion-sensitive analysis, which, for nematogenic 8OCB and 5CB, yielded *α* ≈ 0.25, in agreement with the TCP hypothesis [56,63].

Another ultimate test of the ‘TCP hypothesis’ can be the experimental demonstration of the above-mentioned crossover from a discontinuous phase transition line to a continuous phase transition line. Several experimental studies have suggested that this behaviour can be achieved for the I-N transition under a sufficiently strong external electric field [101,102,103,104,105]. The authors of this report have extensive experience in NDE and Electro-Optic Kerr effect studies, which are inherently associated with strong electric fields. In our opinion, the results reported in refs. [101,102,103,104,105] and related cannot be considered as a definitive validation. First, they involve electric fields up to $\sim 10^{6}\;V/cm$ [101,102,103,104,105]. Such extreme electric fields can introduce numerous parasitic impacts, for instance, related to electrical conductivity and local heating. Second, doubt is related to the visible ‘stretching’ of the I-N transition in the range of even 1–2 K, rather than the strong reduction in the discontinuity metric.

For the authors, another factor can be decisive for the smooth crossover from the line of discontinuous to continuous phase transitions: the length of the rod-like molecule, i.e., the primary structural factor. This may be indicated by the variability of the discontinuity value ${\Delta T}^{*}$ in the homologous series of n-cyanobiphenyls to which 5CB belongs. They range from ~10 K for 14CB to ~0.7 K for 4CB, as shown in the Appendix A. See also refs. [106,107] for similar evidence in other LC homologous series. The rising length of rod-like molecules, matched with the rising ability to undergo a uniaxial ordering, can be a decisive parameter defining the reduction in discontinuity, and a continuous transition can appear below some minimal molecular length. This interpretation supports the recently obtained continuous phase transition, within the experimental error limit, for approaching I-N transition in 5CB on compressing, with a frustrated impact of a small addition of paraelectric BaTiO_3_ nanoparticles [108].

In standard temperature tests for a given LC compound, only one point from such near-TCP line is empirically available. For 5CB, it is associated with a ${\Delta T}^{*}\approx 1.3\;K$ discontinuity.

### 3.3. Discontinuity of the I-N Transition

De Gennes’ question (**Q2**) indicates the unique ‘weakness’ of the I-N transition, which reflects small values of ${\Delta T}^{*}$. It can be precisely determined using properties encountered in Equation (3). This issue was partially discussed at the end of the last subsection. A notable commentary on de Gennes’s concern may be the qualitatively large model estimates of the discontinuity metric. For the basic LdG model, it is ~26 K, and for the alternative Maier–Saupe model, it is even ~42 K [1,4,42,43,44]. Approximate agreement with experiment was obtained by Mukherjee in 1998 [44], assuming the TCP character of the I-N transition.

The discontinuity metric ${\Delta T}^{*}$ value is considered a particularly significant experimental checkpoint for theoretical models [1,2,3,4,5,6,19,42,43,44]. It is also a recognized ‘material’ characteristic of LC materials [4]. However, it is rarely noted that this meaning of ${\Delta T}^{*}$ has some limitations. In nematogenic LC materials, the discontinuity value strongly increases with compression; for example, in 5CB, it can reach ~7 K at a relatively moderate pressure of P~500 MPa [63]. As indicated in refs. [63,106], at least in the homologous series nCB, pressure changes lead to the appearance of the isotropic–nematic–smectic A triple point associated with the only $\Delta T^{*}$ value, which is specific, i.e., only for a given LC material.

### 3.4. Dielectric Constant: Model Explanation

In the Section 2, Drozd-Rzoska’s model analysis [58,63] was presented to describe the pretransitional behaviour of the dielectric constant in the isotropic phase of nematogens. It employed Mistura’s (1974) [78] and Sengers et al.’s (1980) [80] concepts, developed to explain the critical anomaly of the dielectric constant as one approaches the critical consolute point (CCP) in mixtures of limited miscibility.

However, one can also consider the application of the droplet model introduced by Oxtoby for the approximate description of critical effects in liquids (1977) [109], and next implemented to describe anomalies in dielectric properties, namely the dielectric constant, Kerr effect, and NDE in critical binary mixtures (1979) [110]. It led to the following relation of dielectric constant excess change on approaching the critical consolute point in binary mixtures [80,83]:(12)$$\varepsilon \left(T\right)=\varepsilon^{C}+A\left(T-T_{C}\right)+B{\left(T-T_{C}\right)}^{1-\alpha }$$

The Oxtoby model [109,110] considers exclusively the dielectric constant critical anomaly in a system, which arises due to the dielectric constant’s local excess associated with precritical fluctuations, approximated as droplets with a radius described as the correlation length (Equation (11)). Such a picture can also be directly implemented in the isotropic–nematic phase, if system-specific features are taken into account. It includes elements of the inherent uniaxial anisotropy specific to the next nematic phase, as well as a semi-continuous and mean-field type nature of the transition. Following this reasoning, Equation (12) directly transforms into the isotropic phase-related Equation (6), with $T_{C}\;\to \;T^{*}$.

The characteristic feature of continuous/critical phase transition is the appearance of pretransitional fluctuations with local characteristics recalling the next approaching phase. For the isotropic phase of nematogens, it is related to prenematic fluctuations. They have to show the basic feature of the nematic phase, i.e., the uniaxial ordering, whose dominant direction is indicated by the director. It is significant that directors $\vec{n}$ and $\vec{-n}$, indicating the dominant uniaxial ordering, are equivalent [4]. The latter feature directly leads to the approximate ‘cancellation of contribution from permanent dipole moments, if they are parallel to the long molecular axis. Notably, the cancellation does not require intermolecular dipole–dipole coupling and can be purely ‘statistical’. Consequently, the dielectric constant within fluctuations is essentially less than that of the ‘chaotic’ surrounding. A ‘contrast factor’ between fluctuations and their liquid-like surrounding appears. It yields explicit conditions for the above-considered modelling, which recalls the droplet model background.

The same framework excludes the appearance of the dielectric constant pretransitional effect in the isotropic phase of MBBA, specifically with respect to the transverse permanent dipole moment. In the given case, the ‘cancellation’ of the permanent dipole moment has to be absent. There is no ‘contrast factor’, and therefore no pretransitional anomaly.

Regarding the significant difference in the manifestation of pretransitional effects between the isotropic phases of 5CB and 5*CB, it is worth noting that they are isomeric compounds, sharing the same permanent dipole moment with respect to the long molecular axis, but showing a slight structural difference, as illustrated in Figure 1 and Figure 2.

For 5*CB, a steric hindrance associated with the molecular structure distorts the prenematic arrangement within fluctuations, as illustrated in Figure 7. For 5CB, the contribution of permanent dipole moments can be almost totally cancelled. For 5*CB, one can expect a decoupling: $\vec{\mu }={\vec{\mu }}_{\left|\right|}+{\vec{\mu }}_{⊥}$. Only the parallel component can be subject to prenematic cancellation, and then a non-vanishing transverse component of the permanent dipole moment ($\mu_{⊥}$) for the prenematic fluctuations remains. The limited cancellation of the permanent dipole moment leads to a lesser contrast factor in 5*CB than in 5CB. It further leads to a lesser manifestation of dielectric constant pretransitional anomaly in 5*CB than in 5C, as visible in Figure 4 and Table 1.

Dielectric constant scans register the effective changes in this quantity in the total volume of the sample, which in a given case resembles a specific ‘critical colloid’ consisting of the isotropic volume part with chaotically arranged molecules and a part occupied by prenematic (locally ordered) fluctuations: $V_{total}=V_{iso}+V_{fluct}$.

The volume occupied by the fluctuations increases on cooling $V_{fluct}\propto \xi^{3}\propto {\left(T-T^{*}\right)}^{-3}$, which is related to the simultaneous decrease in the $V_{iso}$ contribution. Both contributions to the registered dielectric constant equilibrate at some temperature for which $d\varepsilon \left(T\right)/dT=0$. Below this crossover temperature, in the immediate vicinity of the I-N clearing temperature, the impact of fluctuations dominates.

This is a case of the indirect detection of the impact of critical fluctuations.

### 3.5. Nonlinear Dielectric Effect: Timescale Meaning and the Model Explanation

The Kerr effect (KE), the Cotton–Mouton effect (CME), light scattering (LS), and the nonlinear dielectric effect (NDE) are methods that directly register the response from multimolecular fluctuations. NDE introduces the issue of the interplay between the timescale associated with the scanning electric field employed by the method and the relevant system timescale. The latter is associated with the prenematic fluctuations lifetime, which critically rises on cooling, as shown in Equation (11). For 5CB, it terminates at $\tau_{flukt}\left(T^{C}\right)\sim 10^{-6\;}s$, which is ca. 3 decades less than the molecular primary relaxation time τ [63,74].

KE, CME, and LS are methods employing light for scanning properties. For the ‘red’ light limit, it is related to frequency f~430 THz, and then, the scanning timescale $t_{scan}=1/f=2.3\times 10^{-15\;}s$. Hence, for the mentioned light-related methods, $t_{scan}\ll \tau_{flukt}$ at any temperature in the isotropic liquid phase. NDE detects the system’s properties using a radio-frequency weak electric field, in the kHz–MHz range [66]. It yields the scanning timescale $t_{scan}=1/f$ from ca. 0.1 ms to 0.1 μs. The NDE scanning timescale can coincide with the system timescale ($\tau_{flukt}$) in the isotropic phase.

Experimental evidence for OKE, EKE, CME, and LS shows the (apparent) agreement with the LdG model (Equation (3)), regarding both the temperature behaviour pattern and the method-related amplitude $A_{M}$. The latter includes significant molecular properties, particularly anisotropies of the refractive index $\left(\Delta n\right)$ or dielectric constant (Δε). Following the discussion in the given report, the detection was carried out by OKE, EKE, CME, and LS methods, which were used to ‘observe’ fluctuations as an average from individual molecules constituting the average fluctuation.

The scanning timescale of NDE causes the entire pretransitional (prenematic) fluctuation to be ‘observed’/scanned. In the particular case of low-frequency (LF) NDE, it is possible that $t_{scan}\gg \tau_{flukt}$, and the effective average response from an assembly of fluctuations was tested.

In 1993, Rzoska [40] proposed a model for explaining the mystery of NDE and EKE pretransitional effects in the homogeneous phase of critical binary mixtures. It led to the output equation [40]:(13)$$EKE,\;NDE=C_{M}{\langle {\Delta S}^{2}\rangle }_{V}\chi_{T}\propto {\left(T-T_{C}\right)}^{2\beta }\times {\left(T-T_{C}\right)}^{-\gamma }$$where ${\langle {\Delta S}^{2}\rangle }_{V}$ is the mean of the local ordered parameter fluctuations square and $\chi_{T}$ is order parameter related susceptibility, i.e., the compressibility; $C_{M}$ is the physical magnitude related model constant.

In ref. [41], it was extended for the supercritical domain above the gas–liquid critical point.

Significant are the unique conditions for a ‘standard’ supercritical liquid under a strong electric field [40]:
Initially, spherical/isotropic critical fluctuations become elongated and semi-uniaxial under the strong electric field, so the correlation length becomes uniaxial $\xi =\left(\xi_{\left|\right|},\;\xi_{⊥},\;\xi_{⊥}\;\right)$, and additionally shows a ‘mixed criticality: $\xi_{\left|\right|}\propto {\left(T-T_{C}\right)}^{-\nu \;\approx \;-0.63}$. That is, it follows the standard for critical mixtures non-classical pattern, and $\xi_{⊥}\propto {\left(T-T_{C}\right)}^{-1/2}$, i.e., it follows the mean-field pattern.The different experimental definitions of NDE are significant: ${\Delta \varepsilon }^{E}/E^{2}=\varepsilon \left(E\right)-\varepsilon =\varepsilon_{\left|\right|}-\;\varepsilon$, and EKE,$:\;n_{\left|\right|}-\;n_{⊥}$.


The above model can be extended for the isotropic phase above the I-N transition [40,41]. First, one should consider the substitution $T_{C}\to {\;T}^{*}$ in Equation (13). Next, one should take into account that for deriving Equation (3), only the term $\propto S^{2}$ is taken into account. It means that in the LdG series, given by Equations (1) and (2), other terms can be negligible for the methods encountered in Equation (3). It can be explained by the phase transition discontinuity. The authors of this report indicate that a strong external electric (or magnetic) field can change the tested system, creating a ‘uniaxial’ semi-critical colloid. Consequently, under such conditions, the system may return to its basic mean-field state.

Following Equation (13), it leads to the following dependence in the isotropic liquid phase of nematic LC:(14)$$NDE=C\langle \Delta S^{0}\rangle \langle \Delta S^{f}\rangle \chi_{T}=C\chi_{0}\frac{\langle \Delta S^{0}\rangle \langle \Delta S^{f}\rangle }{T-T^{*}}$$

The mean-field character causes the decoupling of the order parameter fluctuations and the temperature behaviour governed by the susceptibility: $\chi_{T}=\chi_{0}{\left(T-T^{*}\right)}^{-\gamma \;=\;-1}$.

For NDE, local order parameter fluctuations are associated with local dielectric constant fluctuations in the static domain in a hypothetical zero-frequency limit. It is linked to the DC strong electric field, inducing the anisotropy (${\Delta \varepsilon }^{0}$), i.e., zero-frequency extrapolation of the static domain. It reduces the detection of a fluctuation ${\Delta \varepsilon }^{0}$ to a single-molecule-related anisotropy of the dielectric constant. For ${\Delta \varepsilon }^{f}$, it is not possible, and the average single fluctuation features are detected.

Equation (14) appears to resemble the LdG Equation (8). However, the parameters in Equation (14) are related to the ‘whole’ fluctuations, detected via the relevant timescale to the scanning frequency. For LdG Equation (8), there are explicit molecular properties.

Following the information above, one can consider the nature of pretransitional NDE anomalies presented in Figure 5 and Figure 6:

For MBBA, $\langle \Delta S^{0}\rangle \propto \Delta \varepsilon <0$, and one can consider the uniaxial ordering of rod-like molecules with the transverse permanent dipole moment, which allows for a free rotation and then the common orientation under a strong electric field, yielding for the fluctuations $\langle {\Delta S}^{0}\rangle \propto {\Delta \varepsilon }_{fluct}^{0}\sim {\Delta \varepsilon }^{0}<0$ and $\langle {\Delta S}^{f}\rangle \propto {\Delta \varepsilon }_{fluct}^{f}<0$, and then $A_{NDE}>0$, in agreement with Figure 5.
For 5*CB, $\langle \Delta S^{0}\rangle >0$; however, for fluctuations, only the component of the permanent dipole moment parallel to the director can be cancelled due to the prenematic arrangement, and a significant transverse component of the permanent dipole remains (See Figure 7). It can be freely oriented under the strong leading electric field, leading to $\langle {\Delta S}^{f}\rangle <0$ and then $A_{NDE}<0$, i.e., the negative pretransitional anomaly, in agreement with the results presented in Figure 6.For 5CB both $\langle \Delta S^{0}\rangle \propto {\Delta \varepsilon }^{0}>0$ and $\langle {\Delta S}^{f}\rangle \propto {\Delta \varepsilon }_{fluct}^{f}>0$, leading $A_{NDE}>0$, as evidenced in Figure 5.


For a sufficiently low scanning frequency, the averaged response from a few fluctuations is detected, thus eliminating the detection of disturbing impacts, such as pre-Smectic arrangements within fluctuations. It can explain the validity of Equation (13) for the I-SmA or I-SmE transition, for homologous series of n-isothiocyanatobiphenyls and n-thiocyanatobihenyls [106,107]. New experimental insights can facilitate the development of NDE devices with adjustable scanning frequencies.

### 3.6. Distortions from Landau–de Gennes Model Pattern Close to I-N Transition

A significant and still-inconclusive discussion concerns small deviations from the LdG temperature pattern (Equation (3)) that occur in the immediate vicinity of the I-N transition for KE, CME, or LS. They are particularly evident when testing reciprocals of the above properties, via a ‘bending down’ from the expected linear behaviour. It is heuristically explained as the influence of undefined ‘cybotactic groups’ [4,19,30,31,32,33,34,35,36,45,46,47,48,110,111].

For the authors, a possible explanation of this ‘inherent’ distortion is provided when analyzing the phenomenon via the model Equations (13) and (14). The extreme detection frequency for light–related methods, in comparison with the lifetime of fluctuations, causes the detection of fluctuations to be reduced to an apparent average of a single-molecule-related property. The crucial meaning is related to the molecular anisotropy of the refractive index. However, this is an average value seen over the entire uniaxial order within a prenematic fluctuation. The image of the prenematic fluctuation for 5CB in Figure 7 is an idealization. In reality, the molecules are slightly disoriented, and the director indicates only the dominant averaged prenematic direction. Uniaxial ordering improves when approaching the I-N transition, which is also related to the large increase in the number of molecules in the fluctuation. The mentioned methods (KE, CME, and LS), in fact, also detect (anisotropic) fluctuations. Therefore, it can be assumed that the detection using these methods is essentially related to the nondipolar component of dielectric polarization component in the direction indicated by the director $\vec{n}$. As $T^{C}$ approaches, this component increases on $T\to T_{IN}$ due to the improved ordering. The CME, OKE, and EKE amplitudes in Equation (3) also increase in comparison to the behaviour remote from the I-N transition. Consequently, the mentioned ‘bending down’ occurs.

The indirect evidence for this picture could be explained by the anomalous changes in the higher-frequency NDE pretransitional effect shown in the inset in Figure 5.

For NDE, it can be explained by the joint impacts of the measurement frequency and, first and foremost, the imperfect uniaxial ordering, changing the amplitude $A_{NDE}$. On cooling towards $T^{C}$, a registerable effective transverse component for large-enough fluctuations appears, which decreases $A_{NDE}$, and starts the route towards the negative NDE (as for 5*CB). It is the ‘bending up’ that is visible in the inset in Figure 5. However, on approaching the singular temperature, the compressibility critically rises, which, under the strong electric field, improves uniaxiality. The initial trend diminishes, and the ‘bending down’ in the inset in Figure 5 appears. $A_{NDE}$ rises towards the value related to the perfect uniaxial arrangements.

## 4. Materials and Methods

Liquid crystalline compounds were purchased from Fluka (Seelze, Germany), and a few [freezing, vapour removal, melting and heating up to ~120 °C, and then freezing] were carried out. It degassed samples and removed parasitic contaminations. The mesomorphism of the tested compound is given in Section 2.

Broadband dielectric spectroscopy studies were carried out using a Novocontrol impedance analyzer, for frequencies from 1 Hz to ~1 GHz, with the voltage of the measurement (scanning) electric field U=1 V, which yields the optimal resolution of measurements. Samples were placed in a flat-parallel capacitor with the gap d=0.3 mm, which yields $E\approx 33\;V{cm}^{-1}$ for the intensity of the scanning electric field. The capacitor was placed in the Quattro Novocontrol unit for temperature control, enabling stabilization of ~0.02 K. Temperature was monitored using a Pt100 thermistor (UnipressEquipment, Warszawa, Poland), placed within one of the capacitor’s plates.

For the nonlinear dielectric effect (NDE), the single-resonant circuit concept was employed, where a strong electric field pulse is applied to the tested sample, causing a frequency shift. The latter was monitored via a modulation domain analyzer, allowing for frequency versus time scans. The design of the apparatus is given in refs. [66].

The strong electric field was applied in the form of ‘rectangular (DC)’ pulses lasting 1–5 ms, with the voltage changing from 200  V to 1000V. For the applied capacitor, it led to electric field intensities ($E_{strong}$) from $6.7\;kVm^{-1}$ to $33\;kVm^{-1}$. During the experiment, the condition related to Equation (4) ${\Delta \varepsilon }^{E}=\varepsilon \left(E\right)-\varepsilon \left(E\to 0\right)\propto E^{2}$ was permanently tested, and this condition was tested during experiments.

During the experiment, the condition related to Equation (4) ${\Delta \varepsilon }^{E}=\varepsilon \left(E\right)-\varepsilon \left(E\to 0\right)\propto E^{2}$ was permanently tested, and this condition was tested during experiments.

For the weak, radio-frequency-related scanning electric field, the intensity E≈33 Vcm was applied. It was associated with the voltage U=1V applied to the measurement capacitor, which enables an optimal resolution. Note that NDE measurements are associated with extremely small dielectric constant changes $\left(\Delta \varepsilon /\varepsilon \right)\sim 10^{-6}–10^{-8}$, which show the technological challenges of this method [64,65,112,113].

The macroscopic gap of the measurement capacitor is notable, as it avoids surface-related effects that can occur when micrometric gaps are used. The latter are often employed in BDS studies. A notable feature of the NDE technique used in this study is the application of a ‘rectangular’ pulse of the DC electric field, which yields a comparable response from the activated sample. This enables any parasitic impacts that could distort the measurements to be identified during the experiment. These impacts can result from gas bubbles if the tested sample is not satisfactorily degassed or from solid contamination, such as dust particles. These factors all contribute to making the applied NDE measurement concept a reliable experimental solution. Another advantage of the NDE measurement concept is its sensitivity, enabling research in the range of $10^{-20}m^{2}/V^{2}$ to $10^{-15}m^{2}/V^{2}$ and the detection of statistical fluctuations in the polarizability of non-dipole liquids [66]. The only limitation is the lack of commercially available solutions despite the existence of proven advanced solutions, such as those in the laboratory of the authors of the given report [67].

## 5. Conclusions

The isotropic phase of rod-like nematic compounds is a unique experimental model system that provides a significant reference point for theoretical and simulation analyses in *The Physics of Liquid Crystals* and the broader category of *Soft Matter* systems. It is also important for *Critical Phenomena Physics*, due to the emergence of strong, long-range critical-like pretransitional effects despite the discontinuous nature of the I–N transition.

The Landau–de Gennes model (1969/1970) [1,27,28] plays a key role here, as it was inspired by pretransition effects exhibiting the parameterization shown in Equation (3). Numerous monographs and review reports have validated the Landau–de Gennes model by recalling experimental results related to this equation (Equation (3)). However, already in 1974, the model’s creator, Pierre-Gilles de Gennes, in his famous monograph, pointed out troubling problems with the LdG model for the I-N transition, which are recalled as questions **Q1** and **Q2** [1] in the Introduction. Over time, more such issues have emerged and are noted in questions **Q3**–**Q6** in the Introduction. This report presents the first comprehensive response to all these perplexing problems.

Particular attention is paid to evidence associated with the ‘linear’ (dielectric constant) and ‘nonlinear’ (NDE) dielectric studies, where the manifestation of unusual and diverse features of pretransition effects in the isotropic phase is pervasive and crucial for explaining **Q1**–**Q6** quests.

This work demonstrates that a fundamental understanding of various aspects of pretransitional effects, which manifest as apparent deviations from the mean-field description, is possible in the context of *Critical Phenomena Physics* [18,19,79]. However, it is essential to acknowledge factors such as the role of exceptional prenematic and pretransitional fluctuations, the impact of molecular structure on these fluctuations, and the fundamental importance of specific measurement methods, emphasizing the significance of the scanning timescale. This is where the model introduced in refs. [58,63] becomes particularly important, as demonstrated by the discussion of Equations (13) and (14). It enables a consistent description of pretransitional effects, including KE, LS, and NDE. For the latter, it includes the unusual negative NDE shown in Figure 6. It also provides a common interpretation of the pretransitional effects in the isotropic phase of nematogens and in the supercritical domains near the critical consolute and gas–liquid critical points. Finally, it answers question **Q1**, which particularly troubled de Gennes.

The report shows that it is possible to provide a coherent explanation for questions **Q1**–**Q6**, indicating puzzling problems related to the isotropic phase of nematogens.

A significant part of the report constitutes the results of focused dielectric constant and nonlinear dielectric effect studies in the isotropic phase of selected LC compounds. Notably, a similar, critical-like pattern has been evidenced for nanocolloidal systems composed of nCB liquid crystals and nanoparticles, such as C_60_ fullerenes and BaTiO_3_. These reports also cover LC mesophases and the solid phase, and in each case, the dominant impact of pretransitional fluctuations was demonstrated. These results are particularly important when considering the emerging applications of such materials in innovative renewable energy devices. The report does not address issues related to the dynamics of nematogens in the isotropic phase, which remain a significant challenge for model analysis within Landau–de Gennes theory. However, it provides an extremely important reference for glass transition physics. In this context, the universal critical-type characteristics of changes in the primary relaxation time are worth mentioning, as these describe both dynamics in the isotropic phase as it approaches the nematic phase and the glass temperature in supercooled liquids, as demonstrated by Drozd-Rzoska in ref. [26]:$$\tau \left(T\right)=C_{\Omega }{\left(t^{-1}exp\left(t\right)\right)}^{\Omega }=C_{\Omega }{\left(\frac{T-T^{+}}{T}\right)}^{-\Omega }exp\left(\Omega \frac{T-T^{+}}{T}\right)$$
where $t=\left(T-T^{+}\right)/T$ is the relative distance for the extrapolated singular temperature.

Numerous reports have discussed the implementation of the Landau–de Gennes model for liquid crystalline systems, particularly in the context of the isotropic liquid phase. The latter represents the initial consideration of the system in the history of Model I, which remains its hallmark reference. However, none of these reports have yet addressed a fundamental issue from the perspective of the scientific method [114]: coherently addressing the numerous problems that arise when comparing fused experimental evidence with the basic LdG model. This report is the first of its kind, demonstrating that many experimental phenomena that are surprising in the context of the LdG model can be coherently explained using the model. However, further challenges arise, e.g., related to the issue of dynamic phenomena in the isotropic phase, as indicated above.

The question arises as to whether the model explanations proposed and discussed in the given report fall within or beyond the framework of the Landau–de Gennes model. In our opinion, they are a crucial supplementation to the canonical Landau–de Gennes model and should be considered in any of its future developments, particularly in light of the clear experimental evidence.

## Figures and Tables

**Figure 1 ijms-26-09849-f001:**
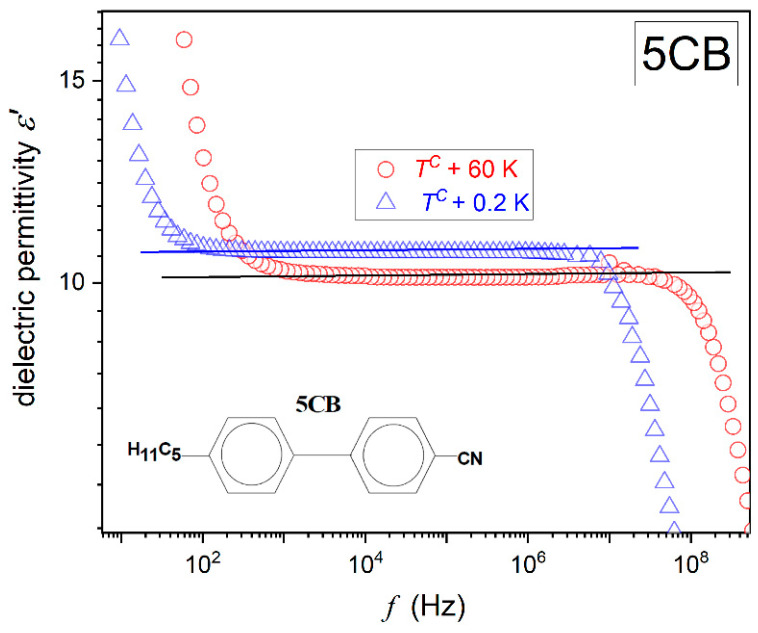
The real part of the dielectric permittivity in the isotropic liquid phase of pentylcyanobiphenyl (5CB), in the immediate vicinity and well remote from the clearing temperature $T^{C}$. The horizontal lines indicate the stationary domain, defining the dielectric constant, i.e., $\varepsilon^{\prime }\left(f\right)=\varepsilon =const.$ The plot includes the schematic presentation of the 5CB molecular structure.

**Figure 2 ijms-26-09849-f002:**
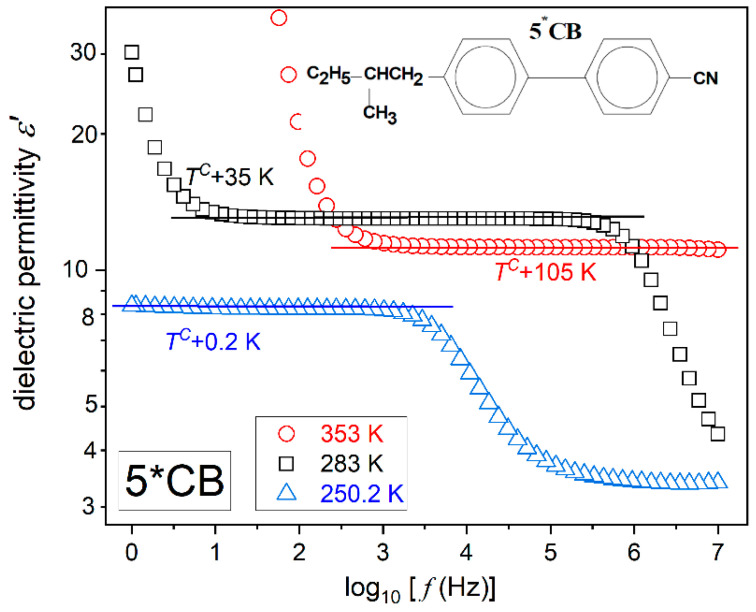
The real part of the dielectric permittivity in the isotropic liquid phase of isopentylcyanobiphenyl (5*CB), in the immediate vicinity and well remote from the clearing temperature $T^{C}$. The horizontal lines indicate the stationary domain, defining the dielectric constant, i.e., $\varepsilon^{\prime }\left(f\right)=\varepsilon =const.$ The plot includes the schematic presentation of the 5*CB molecular structure.

**Figure 3 ijms-26-09849-f003:**
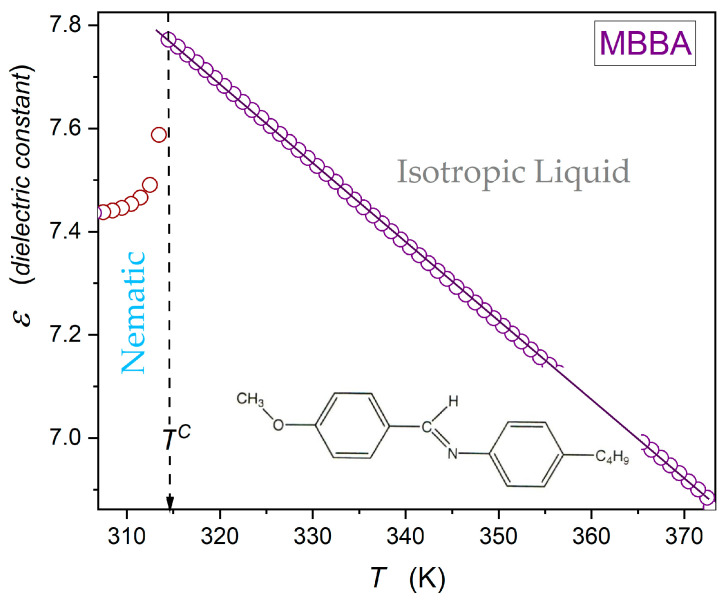
Linear changes in the dielectric constant for the isotropic liquid phase of MBBA, the nematogenic LC compound with the permanent dipole moment approximately parallel to the long molecular axis.

**Figure 4 ijms-26-09849-f004:**
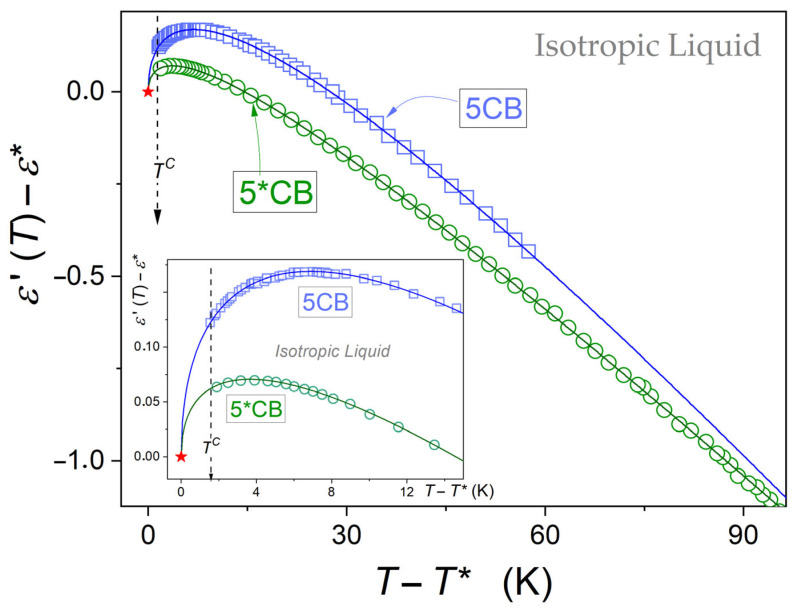
Pretransitional changes in the dielectric constant in the isotropic liquid phase of 5CB and 5*CB. Results are presented in the normalized scales with respect to $\left(\varepsilon^{*},\;T^{*}\right)$ parameters, indicated by a star in red. Experimental data are parameterized using Equation (6), with the parameters listed in Table 1. The inset shows the focused behaviour close to the clearing temperature.

**Figure 5 ijms-26-09849-f005:**
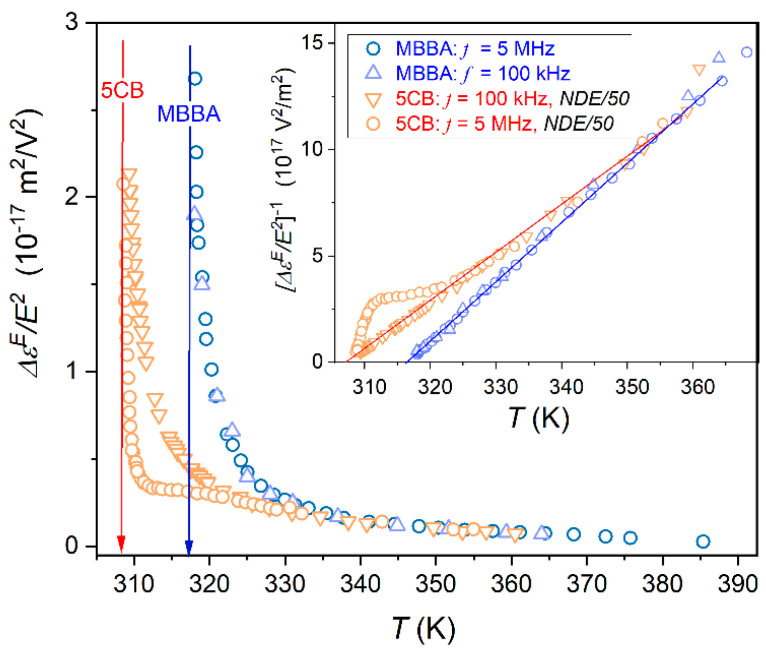
NDE pretransitional effect in the isotropic liquid phase of MBBA and 5CB, for two scanning frequencies given in the plot. The inset shows the reciprocals of these data, with linear changes corresponding to the Landau–de Gennes model temperature behaviour: Equations (3) and (8). The obtained values of the discontinuity metric are as follows: ${\Delta T}^{*}\approx 1.4\;K$ for 5CB and ${\Delta T}^{*}\approx 0.8\;K$ for MBBA. For comments on the meaning of the applied scanning frequencies and how they interplay with the timescale introduced by pretransitional fluctuations, see Section 3.5 below.

**Figure 6 ijms-26-09849-f006:**
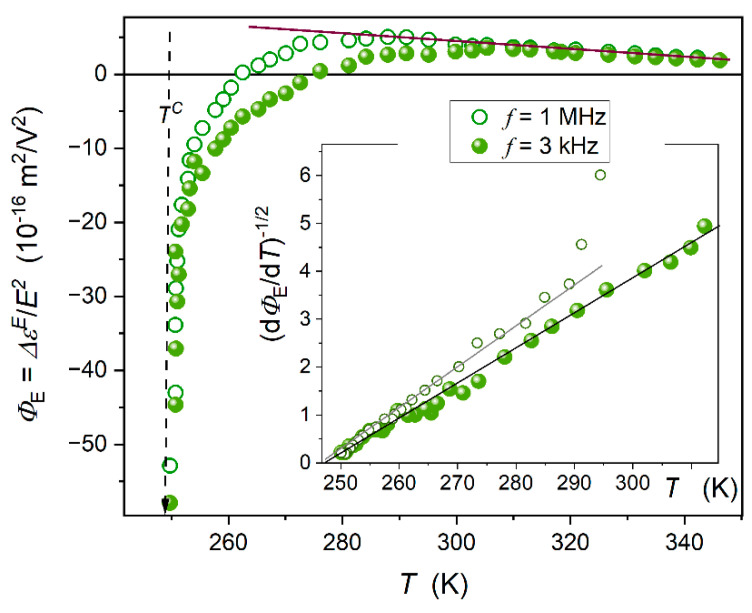
Nonlinear dielectric effect ($\Phi_{E}={\Delta \varepsilon }^{E}/E^{2}$) in the isotropic liquid phase of isopentylcyanobiphenyl (5*CB) for two scanning frequencies, f=3 kHz and f=1 MHz of the weak (scanning) electric field. The inset is for the linearized, derivative-based analysis introduced in Equation (9). It validates the temperature pattern ${\Delta \varepsilon }^{E}/E^{2}\propto 1/\left(T-T^{*}\right)$, despite the unique ‘negative anomaly’. The obtained value of the discontinuity metric ${\Delta T}^{*}\approx 1.7\;K$. The vertical dashed arrow shows I-N* clearing temperature. The line in brown, remote from the clearing temperature, is related to the Piekara Equation (10). For comments on the meaning of the applied scanning frequencies and how they interplay with the timescale introduced by pretransitional fluctuations, see Section 3.5 of the Discussion below.

**Figure 7 ijms-26-09849-f007:**
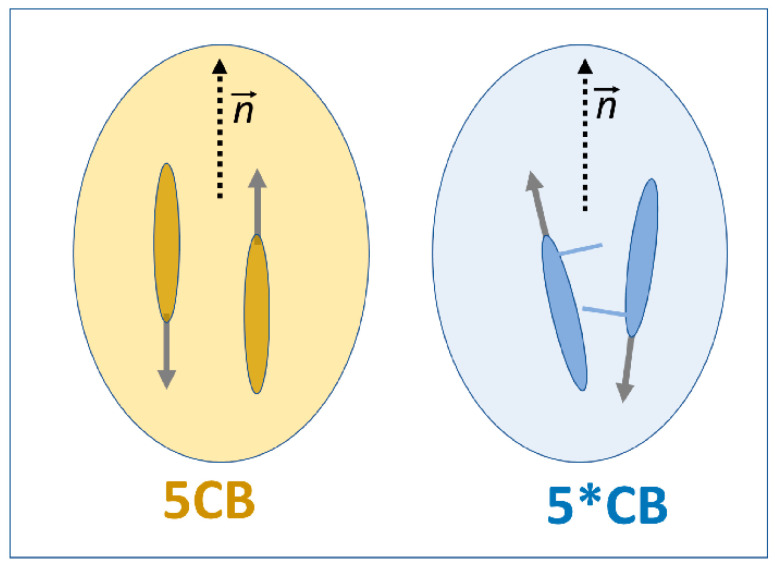
The schematic presentation of molecular arrangements within prenematic fluctuations for 5CB and 5*CB. Elongated ellipsoids represent the rod-like part of molecules. Solid arrows indicate permanent dipole moments. For 5*CB, the structural hindrance is also schematically shown. The dotted arrows are for ‘directors’, which indicate preferred directions of the prenematic arrangement.

**Table 1 ijms-26-09849-t001:** Values of parameters describing behaviour in the isotropic liquid phase for tested LC compounds with longitudinal (5CB and 5*CB), recalling Figure 3 and Equation (6).

LC	$A_{\varepsilon }$ (K^−1^)	$B_{\varepsilon }$ (K^−0.5^)	$T_{cross.}$ $\;\left(d\varepsilon /dT\;=\;0\right)$: (K)	α
5CB	−0.025_0_	0.129_0_	8.2	1/2
5*CB	−0.019_4_	0.074_2_	3.7	1/2

## Data Availability

All data are available directly from the authors following a reasonable request. They are also deposited in the public open-access REPOD database (Poland).

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
