# Peer review of "Landau–de Gennes Model for the Isotropic Phase of Nematogens: The Experimental Evidence Challenge"

_ijms, 2025, doi:10.3390/ijms26209849_

Round 1

Reviewer 1 Report

Comments and Suggestions for Authors

The core of the paper focuses on the experimental challenges of the Landau-de Gennes (LdG) model. It proposes a coherent explanation for the six major experimental issues (Q1-Q6). The paper repeatedly mentions that "it is necessary to determine whether the explanation goes beyond the scope of the LdG model," yet it never clearly specifies whether the explanation falls within the framework of the LdG model. This makes it difficult for readers to judge whether the new explanation serves as a supplement and improvement to the LdG model or constructs a new framework independent of the LdG model. The authors need to clarify this aspect. Additionally, there are some other issues worthy of attention:

  1. The experiment selected three rod-like nematic liquid crystal compounds (5CB, 5CB, MBBA). What is the selection principle? Do the research results obtained from them have universality?
  2. The test data in Figure 5 and Figure 6 use different test frequencies. What is the reason for this? What impact does the test frequency have on the NDE (nonlinear dielectric effect) pretransitional effect of different materials?
  3. The voltage used in the broadband dielectric spectroscopy studies is 1V, while the voltage employed in the nonlinear dielectric effect (NDE) studies ranges from 200V to 1000V. What is the rationale behind this voltage selection? Has consideration been given to the different patterns of experimental results under the influence of different voltages?
  4. The references are relatively outdated, and it is necessary to update the relevant research literature from the past two years.

Author Response

Attached please find the corrected report ‘ Landau – de Gennes model for the isotropic phase of nematogens: the experimental evidence challenge’, ref. ijms-3894823.

Generally, opinions of reviewers are positive, and address explanation recalling some basics of liquid crystals physics with respect to implementations of ‘linear’ (dielectric constant) and ‘nonlinear’ (NDE) dielectric scanning methods. It shows that the report requires supplementary explanations for readers. We did it, explicitly following the Reviewers' questions and suggestions.

We are grateful to reviewers for their efforts.

Reviewer #1

General Comment: The general criticism is only formal:

The paper repeatedly mentions that "it is necessary to determine whether the explanation goes beyond the scope of the LdG model,"

Response: This statement appeared only 2x in the submitted manuscript:

  • at the end of the Abstract and (ii) at the end of Conclusions.

Now, there is no such statement in the Abstract, and in the Conclusions a short comment is added. It means this statement appears only 1x.

Note additional references, introduced to response to Reviewers comments.

Specific Comments

  1. Reviewer #1: ‘The experiment selected three rod-like nematic liquid crystal compounds (5CB, 5CB, MBBA). What is the selection principle? Do the research results obtained from them have universality?’

Response: This issue was/is explained at the beginning of the Results section (the last 11 lines on page 3 and the first 11 pages in page 4). Nevertheless, to conclude this point explicitly, the following text has been added (lines 12-19, page 4), namely:

In conclusion, the tested compounds are characterised by qualitatively different positions of the permanent dipole moment (5CB, MBBA) and the presence of structural steric hindrance (5*CB). These factors have a significant impact on the form of pretransitional effects in the isotropic liquid phase, as noted in the literature and as presented in the results below. This is a unique situation, as the standard properties studied in the isotropic phase, such as compressibility (usually tested via KE or IR measurements) and heat capacity (specific heat), consistently exhibit the same pretransitional patterns. This includes 5CB, 5*CB, and MBBA, tested in the given report.

  1. Reviewer #1: ‘The test data in Figure 5 and Figure 6 use different test frequencies. What is the reason for this? What impact does the test frequency have on the NDE (nonlinear dielectric effect) pretransitional effect of different materials?

Response: Both in dielectric constant and NDE (the strong electric field-related static extension), one should follow the static domain. But for 5*CB it strongly shifts to lower frequencies, as shown in Figures 1 and 2 (so-called masterplot). Additionally, for NDE – which directly tests pretransitional fluctuations - essential is the time-scale associated with these fluctuations, in comparison to the measurement scanning frequency.

The significance and the impact of the interplays between these time scales was/is in-depth commented on in Subsection 3.5.

To clarify the situation for readers in the Results section and to guide reading in the subsequent ‘Discussion’ section, the following comment has been added in captions of Figures 5 and 6:

For comments on the meaning of the applied scanning frequencies and how they interplay with the timescale introduced by pretransitional fluctuations, see section 3.5 of the Discussion below.

  1. Reviewer #1: ‘The voltage used in the broadband dielectric spectroscopy studies is 1V, while the voltage employed in the nonlinear dielectric effect (NDE) studies ranges from 200V to 1000V. What is the rationale behind this voltage selection? Has consideration been given to the different patterns of experimental results under the influence of different voltages’

Response: for BDS, i.e., dielectric constant, related studies it is/was explicitly stated ‘the voltage of the measurement (scanning) electric field , which yields the optimal resolution of measurements. Maybe here the link to parameters at  Novocontrol GmbH would be optimal, but such a commercial link is not proper/allowed. The same is for NDE.

The reviewer, in the comments ‘mixes’ the measurement (U=1, radio frequency) and the DC strong electric field (200 V – 1000V). The first scans properties, the second creates anisotropy.

All these is explicitly explained in the Methods section, which is given at the end of the report – according to the required structure of IJMS.

Nevertheless, we decided slightly rearranged NDE – related part of the Methods section, to avoid misunderstanding for readers beyond the nonlinear dielectric spectroscopy topic.

  1. Reviewer #1: ‘The references are relatively outdated, and it is necessary to update the relevant research literature from the past two years.

Response:  Please note 10 new references, Half of them are related to the last 2 years.

Reviewer 2 Report

Comments and Suggestions for Authors

The manuscript describes Landau – de Gennes model for the isotropic phase of nematic substances and focuses on dielectric response dependence on temperature of polar nematic organic substances. This dependence is well interpreted and rationalized. However, I cannot recommend this manuscript in the present form, first, because it is written for physicist – not chemists or interdisciplinary specialist, who make up the IJMS audience.

I habe some minor suggestion to improve.

  1. Why the authors focus only on the dielectric permittivity and do not make the bridge to the polarizability. These are interdependent quantities connected with Clausius – Mossotti equation. Furthermore, there is another equation of de Gennes [P.G. de Gennes, Wetting: statics and dynamics, Rev. Mod. Phys. 1985, 57, 827] uniting interfacial wetting parameter (contact angle Q) with polarizabilities (a) of the interaction phases: cos Q = 2*a1/a2 – 1.
  2. I recommend the authors enhancing the discussion of possible practical use of their finding beyond the liquid-crystalline physical chemistry. Please, mention its applicability in the physical chemistry of organic solar cells based on nanosized fullerene molecules. Indeed, here the crucial effect of the dielectric permittivity on the key output parameters of photovoltaic devices is discussed (See: Koster et al., Pathways to a new efficiency regime for organic solar cells, Adv. Energy Mater. 2012, 2, 1246, https://doi.org/10.1002/aenm.201200103].
  3. The role of anisotropy of polarizability and anisotropy of dielectric permittivity of the molecules in the photovoltaic performances is also discussed in [Sabirov et al., The C70 fullerene adducts with low anisotropy of polarizability are more efficient electron acceptors for organic solar cells. The minimum anisotropy hypothesis for efficient isomer-free fullerene-adduct photovoltaics, Journal of Physical Chemistry C 2016, 120, 24667, https://doi.org/10.1021/acs.jpcc.6b09341].
  4. The abstract does not provide the information about the novelty of the findings of the present piece of work as compared with the previous ones. It only states the logic of the study. Please, correct.
  5. The manuscript is written somewhat untidily, e.g. some of the quantity names are capitalized as the others are not. This must be unified. Some terms are uncommon, e.g. nondipolar liquid (nonpolar liquid?). Chemical structures of nCB compounds must be shown in Appendix or Supplementary Materials.
  6. Do the authors observe hysteresis in the measurements shown in Figures 1 and 2 (real permittivity as function of frequency)?
  7. The authors consider phase transition and fluctuations with no invoking entropy. Why?
  8. The theoretical constructions and interpretations of the experiments were done for the selected class of organic compounds (cyanoalkylbiphenyls). Are the findings transferrable to other compounds and extendable to general case? Please, substantiate.

Author Response

Attached please find the corrected report ‘ Landau – de Gennes model for the isotropic phase of nematogens: the experimental evidence challenge’, ref. ijms-3894823.

Generally, opinions of reviewers are positive, and address explanation recalling some basics of liquid crystals physics with respect to implementations of ‘linear’ (dielectric constant) and ‘nonlinear’ (NDE) dielectric scanning methods. It shows that the report requires supplementary explanations for readers. We did it, explicitly following the Reviewers' questions and suggestions.

We are grateful to reviewers for their efforts.

Reviewer #2

General comment: ‘The manuscript…….is well interpreted and rationalized. However, I cannot recommend this manuscript in the present form, first, because it is written for physicist – not chemists or interdisciplinary specialist, who make up the IJMS audience.

 Response: Thank you very much for the nice comment. The mentioned interdisciplinary significance is in-depth commented on in the extended beginning of the Introduction (Page 1) and continued on Page 2.

Note additional references, introduced to response to Reviewers comments.

Specific comments:

  1. Reviewer #2: ‘Why the authors focus only on the dielectric permittivity and do not make the bridge to the polarizability. These are interdependent quantities connected with Clausius – Mossotti equation. Furthermore, there is another equation of de Gennes [P.G. de Gennes, Wetting: statics and dynamics, Rev. Mod. Phys. 1985, 57, 827] uniting interfacial wetting parameter (contact angle Q) with polarizabilities (a) of the interaction phases: cos Q = 2*a1/a2 – 1.

Response: direct measurement of polarizability is significant and introduces the cognitive added value in ferroelectric materials (solid or liquid) where the polarizability is also the order parameter. This is not the case with the isotropic liquid of standard LC materials. Here essential is dielectric constant (epsilon = dP/dE) and just for this properties is related to the order parameter. This is the reason why in the given report and in all references we now, there are no direct measurements of electric polarizability in frames of phenomenon discussed in the given report.

Please also note that the Clausius–Mosotti local field is not adequate for the isotropic liquid phase and similar dielectric materials. This is the central topic of Dielectric Physics, considered first by Debye and next Onsager, Kirkwood, and Froelich

Next, the indicated reference in Rev. Mod. Phys. is related to the liquid-solid interface, which is stated in its first sentence. This topic/problem is beyond the issues discussed in the given report.

I know (from personal discussions with the Gennes) that this report is associated with his interests in glues, which he developed in the late 1980s and presented explicitly at the Societa Fisica Italiana general meeting just after obtaining the Nobel Prize.

  1. Reviewer #2: ‘I recommend the authors enhancing the discussion of possible practical use of their finding beyond the liquid-crystalline physical chemistry. Please, mention its applicability in the physical chemistry of organic solar cells based on nanosized fullerene molecules. Indeed, here the crucial effect of the dielectric permittivity on the key output parameters of photovoltaic devices is discussed (See: Koster et al., Pathways to a new efficiency regime for organic solar cells, Adv. Energy Mater. 2012, 2, 1246, https://doi.org/10.1002/aenm.201200103].

Response: In recent years, the authors have significantly focused on nanocolloids, liquid crystal–nanoparticles, also related to fullerenes, with a focus on the interplay with pretransitional fluctuation, which often appears to be decisive. Soon, the next report will be submitted to Nanomaterials (E7 LC mixture + fullerenes).

We did not recalled this topic in the manuscript,  because we were not convinced if it is within the range of problems discussed. Nevertheless, following the reviewer comment, we introduced some indicative discussion in the Conclusions section.

  1. Reviewer #2: ‘ The role of anisotropy of polarizability and anisotropy of dielectric permittivity of the molecules in the photovoltaic performances is also discussed in [Sabirov et al., The C70 fullerene adducts with low anisotropy of polarizability are more efficient electron acceptors for organic solar cells. The minimum anisotropy hypothesis for efficient isomer-free fullerene-adduct photovoltaics, Journal of Physical Chemistry C 2016, 120, 24667, https://doi.org/10.1021/acs.jpcc.6b09341].

Response: This is the same problem as above, and addresses issues beyond the topic and target of the given report. Notwithstanding, this problem is now mentioned in the Conclusions.

  1. Reviewer #2: ‘The abstract does not provide the information about the novelty of the findings of the present piece of work as compared with the previous ones. It only states the logic of the study. Please, correct.

Response: It has been improved.

  1. Reviewer #2: ‘ The manuscript is somewhat untidy, for example, some of the quantity names are capitalised while others are not. This must be standardised. Some terms are uncommon; for example, 'nondipolar liquid'? The chemical structures of nCB compounds should be included in the appendix or supplementary materials.’

Response: The paper has been supplemented by the link to nanocolloidal systems at the end of Conclusions. For other LC systems, namely different series of LC material and transitions to I-SmA and I-SmE phases  were/are mentioned on page 17: starting from line 6 down.

 The term 'non-dipolar liquids' is standard in the field of dielectric physics and refers to materials with a permanent dipole moment. In my opinion, replacing it with 'polar', which may have a broader meaning, could lead to misunderstandings in the context of the problem discussed in this paper.

The molecular structures of 5CB and 5*CB are given explicitly in Figures 1 and 2. For MBBA, a clear reference is made to our recent work (May 2025). However, we have also included this structure in Figure 3.

Therefore, the revised manuscript now contains all the necessary information for interpreting the results.

  1. Reviewer #2: ‘ Do the authors observe hysteresis in the measurements shown in Figures 1 and 2 (real permittivity as function of frequency)?

Response: No. There are also no fundamental bases/references for such a in the isotropic phase of nematogens.

  1. Reviewer #2: ‘ The authors consider phase transition and fluctuations with no invoking entropy. Why?

Response:  Entropy changes underlie the behaviour of heat capacity, which is not the topic of this report. It is also not tested in the isotropic phase because the basic integrative routine for obtaining entropy from heat capacity studies introduces biasing terms. The calculation of the entropy component associated that can be derived from dielectric constant values. It was done by Jadzyn et al. [J. Phys. Chem. 112 (2008), and did not show a relevant/significant link to the topic discussed in the given report.

Nevertheless, the reviewer’s comment is worth developing in further studies, beyond the given report.

  1. Reviewer #2: ‘The theoretical constructions and interpretations of the experiments were done for the selected class of organic compounds (cyanoalkylbiphenyls). Are the findings transferrable to other compounds and extendable to general case? Please, substantiate.

Response:   Of course, YES. This was done to avoid bias from the primary target of this report. Now, this topic is recalled in the Conclusions and particularly on Page 17, six line down.

Round 2

Reviewer 1 Report

Comments and Suggestions for Authors

The authors have basically answered all my questions. If the experimental results are obtained under specific conditions, please indicate the limiting conditions in the corresponding part of the text. At the same time, the research conclusions need to be concise, highlighting the important research findings.

Author Response

  1. Unique experimental conditions description - which  are mainly related to the strong electric field NDE tetst - have been further described in the Methods section. See the supplementations given in lines 748 - 760
  2. The results of the report have been additionaly emphasized, as suggested, in the end of the Conclusions sections. See lines 815-828. 
  3. The latter was associated with an additional reference [116].
  4. The ultimate language 'cleaning' has been also carried out. 

Reviewer 2 Report

Comments and Suggestions for Authors

I have learnt the changes mady by the authors when revising. Most of scientific questions have been resolved; English grammar and notations have been corrected. Thus, the manuscript has been sufficiently improved and now it deserves the publication in IJMS. 

Author Response

Following comments I have undrestood that the report has been accepted. 

Nevertheless, I have made the final-final language test.